# Long Covid stigma: Estimating burden and validating scale in a UK-based sample

**Marija Pantelic**[1,2]*, **Nida Ziauddeen**[3,4], **Mark Boyes**[5,6], **Margaret E. O'Hara**[7], **Claire Hastie**[7], **Nisreen A. Alwan**[3,4,8]*

**1** Brighton and Sussex Medical School, University of Sussex, Falmer, United Kingdom, **2** Department of Social Policy and Intervention, University of Oxford, Oxford, United Kingdom, **3** School of Primary Care, Population Sciences and Medical Education, Faculty of Medicine, University of Southampton, Southampton, United Kingdom, **4** NIHR Applied Research Collaboration Wessex, Southampton, United Kingdom, **5** School of Population Health, Faculty of Health Sciences, Curtin University, Perth, Australia, **6** Curtin enAble Institute, Faculty of Health Sciences, Curtin University, Perth, Australia, **7** Patient contributor, Long Covid Support https://www.longcovid.org, **8** NIHR Southampton Biomedical Research Centre, University of Southampton and University Hospital Southampton NHS Foundation Trust, Southampton, United Kingdom

* M.Pantelic@bsms.ac.uk (MP); N.A.Alwan@soton.ac.uk (NAA)

**Data Availability Statement:** The survey data is available on request provided ethics committee approval for sharing the anonymized data is granted. To request access conditional on approval, please email "rgoinfo@soton.ac.uk," the

## Abstract

### Background

Stigma can be experienced as perceived or actual disqualification from social and institutional acceptance on the basis of one or more physical, behavioural or other attributes deemed to be undesirable. Long Covid is a predominantly multisystem condition that occurs in people with a history of SARSCoV2 infection, often resulting in functional disability. This study aimed to develop and validate a Long Covid Stigma Scale (LCSS); and to quantify the burden of Long Covid stigma.

### Methods

Data from the follow-up of a co-produced community-based Long Covid online survey using convenience non-probability sampling was used. Thirteen questions on stigma were designed to develop the LCSS capturing three domains–enacted (overt experiences of discrimination), internalised (internalising negative associations with Long Covid and accepting them as self-applicable) and anticipated (expectation of bias/poor treatment by others) stigma. Confirmatory factor analysis tested whether LCSS consisted of the three hypothesised domains. Model fit was assessed and prevalence was calculated.

### Results

966 UK-based participants responded (888 for stigma questions), with mean age 48 years (SD: 10.7) and 85% female. Factor loadings for enacted stigma were 0.70–0.86, internalised 0.75–0.84, anticipated 0.58–0.87, and model fit was good. The prevalence of experiencing stigma at least 'sometimes' and 'often/always' was 95% and 76% respectively. Anticipated and internalised stigma were more frequently experienced than enacted stigma. Those who reported having a clinical diagnosis of Long Covid had higher stigma prevalence than those without.

University of Southampton's Research Integrity and Governance team.

**Funding:** The study received no specific funding.

**Competing interests:** NAA and MP are co-investigators on the NIHR-funded STIMULATE ICP study (https://www.stimulate-icp.org). NAA is a co-investigator on the HI-COVE study (https://blog.westminster.ac.uk/hicovestudy) and has contributed in an advisory capacity to WHO and the EU Commission's Expert Panel on effective ways of investing in health meetings in relation to post-COVID-19 condition. MB is supported by the National Health and Medical Research Council, Australia (Investigator Grant 1173043). NZ is supported by NIHR Applied Research Collaboration Wessex. The views expressed are those of the authors' and not necessarily those of the NIHR or the Department of Health and Social Care. This does not alter our adherence to PLOS ONE policies on sharing data and materials.

## Conclusion

This study establishes a scale to measure Long Covid stigma and highlights common experiences of stigma in people living with Long Covid.

## Introduction

Stigma is a process through which individuals are 'disqualified from full social acceptance' due to a physical, health and/or behavioural attribute deemed to be 'deeply discrediting' [1]. The detriment of stigma on both patient and health service delivery outcomes has been well-documented globally. Stigma–and the resultant fears of being ostracised or discredited–drives people underground and away from health services and contributes to psychological distress, thus compromising long-term physical health outcomes [2–5].

Long Covid is a predominantly multisystem condition that occurs in people following SARSCoV2 infection, often resulting in prolonged ill health and functional disability limiting their day-to-day activities including work, learning, care, and leisure activities [6]. In the UK alone, it is estimated that 1.8 million people currently have Long Covid for a duration of at least 4 weeks, out of those 791,000 have had it for at least one year and 235,000 for at least 2 years [7]. Emerging testimonies illustrate profound stigmas experienced by people living with Long Covid [8–11].

People living with poorly understood and managed health conditions such as Long Covid may experience stigma through three main mechanisms [2]. Enacted stigma refers to direct overt experiences of discrimination, whereby individuals are treated unfairly due to their health condition [12]. Internalised stigma occurs when people adopt negative associations with a health condition and accept them to be true and applicable to themselves; this is characterised by feelings of shame and worthlessness [13]. Anticipated stigma is the expectation of bias or poor treatment by others [14]. These mechanisms can occur independently from each other–for example, a person may anticipate stigma, decide against disclosing their health condition or seeking treatment and therefore avoid enacted stigma. Nonetheless, all three stigma mechanisms may undermine people's emotional wellbeing, health seeking behaviours, and physical and mental health outcomes [3].

Research on Long Covid stigma is still in nascent stages. To the best of our knowledge, the extent of stigma experienced by people living with Long Covid has not been quantified using a specific scale, and evidence-based stigma reduction strategies are lacking. One of the reasons for this is a lack of a validated quantitative scale for measuring Long Covid stigma.

This study has two aims: 1) To validate a new Long Covid Stigma Scale (LCSS) and establish its basic psychometric properties; and 2) To quantify the burden of stigma, with its three domains of enacted, anticipated and internalised, as experienced in a UK community-based sample of people with lived experience of Long Covid.

## Methods

Data from the follow-up of a Long Covid online survey was used [15]. The baseline survey, administered in November 2020 (n = 2550), utilised convenience non-probability sampling via social media to ensure recruitment of a community sample of people who identify as living with Long Covid [15].

The survey was restricted to adults aged 18 years or over who had COVID-19 (confirmed or suspected) and were not hospitalised for the treatment of COVID-19 in the first two weeks

of experiencing symptoms. The original scope was decided as very little research was being conducted at the time to describe prolonged illness following non-hospitalised SARSCoV2 infection and most research was focusing on those who were admitted to the hospital during their acute COVID19 illness. People who experienced COVID-19 symptoms and did not require hospital admission during Spring 2020 in the UK largely did not have access to testing, leaving many people with undiagnosed COVID-19 and Long Covid. To enable inclusion of this population, the survey was open to people who did not have lab confirmation of their infection, but had suspected or clinically diagnosed COVID-19.

Participants provided written informed consent (recorded digitally on the survey platforms). Participants had to consent separately to participating in the baseline and follow-up survey before they could access the questionnaire. Responses were anonymous in the baseline survey, but participants who were willing to be contacted for a follow-up survey were asked to consent to future contact and provide contact details. 2210 (86.7%) individuals consented to future contact. One year from the baseline survey, in November 2021, the same participants were invited to complete the follow-up survey. The survey was not open to new participants who did not take part in the baseline survey. Ethical approval was granted by the University of Southampton Faculty of Medicine Ethics Committee (ID 61434).

## Measures

The survey was co-produced working with public contributors (MEO, CH), who have lived experience of and provide peer support to others with Long Covid [16]. NAA also had lived experience of Long Covid. Public contributor members of the COVID-19 Research Involvement Group provided feedback on early versions of the questionnaire which was amended accordingly. Qualtrics was used as the platform for the follow-up following feedback from the baseline survey about user-friendliness.

Questions included demographic information, ability to work, current employment status, pattern of illness and impact on health, symptoms that have remained over the longer-term course, clinical diagnosis of Long Covid and other conditions, vaccination, and experiences of stigma. In this paper, we focus on reporting the stigma-related results of the follow-up survey.

## Long Covid Stigma Scale (LCSS)

For the follow up survey, thirteen questions on stigma were designed following the Health Stigma and Discrimination Framework [2] to capture three dimensions/domains: enacted (5 items), internalised (4 items), and anticipated (4 items). Questions were based on existing scales related to other stigmatised chronic conditions including Myalgic Encephalomyelitis/ Chronic fatigue syndrome (ME/CFS) and HIV [17–20], emerging qualitative evidence on Long Covid stigma [9], and co-production with and feedback from people living with Long Covid. Response options were offered on a 5-point Likert scale: never, rarely, sometimes, often, and always, coded 0–4.

## Hypothesised correlates for assessing concurrent criterion validity

For the purposes of assessing concurrent criterion validity, the follow-up survey also incorporated measures of depressive symptoms and disclosure concerns as these constructs have been consistently positively associated with health-related stigma [21–23]. Depressive symptoms were measured via the eight-item Patient Health Questionnaire (PHQ-8), scored on a four-point Likert-type scale ranging from not at all (0) to nearly every day (3), usually used to assess current depressive symptoms in population-based studies [24–26]. The total score was calculated as the sum of the individual item responses, with higher scores indicating greater

depressive symptom severity. Disclosure concerns were measured via two items about being careful about disclosing information, or regretting having told people about their Long Covid [18]. Responses range from never (0) to always (4). The two items were combined into a total score, with higher scores indicating higher disclosure concerns.

**Analysis strategy.** Analyses presented in this paper focus on the UK follow-up survey sample. Evidence on other health conditions consistently suggests that stigma is sensitive to cultural and geographic contexts [27, 28]. Analyses based on the global follow-up survey sample are provided in S1–S5 Tables.

Data were analysed using Stata v17 [29]. A descriptive preparatory phase examined frequencies, means, and standard deviations to capture the basic sociodemographic characteristics. This was followed by two phases.

**Phase 1: Establishing the psychometric properties of the LCSS.** Confirmatory factor analysis (CFA) tested whether LCSS consisted of the three hypothesised factors (stigma domains): anticipated, enacted and internalised stigma. Model fit was assessed via multiple goodness-of-fit measures. Comparative Fit Index (CFI) and Tucker Lewis Index (TLI) above 0.95 [30, 31], and Root Mean Square Error of Approximation (RMSEA) and standardised root mean-square residual (SRMR) values below 0.05 indicated good model fit [32]. $\chi^2$ was not used to assess goodness-of-fit as it is sensitive to sample size [33]. However, we report $\chi^2$/df as an additional measure of model fit, with values between 1 and 5 indicating good model fit [34, 35]. Internal consistency (Cronbach's α) assessed reliability. Concurrent criterion validity was evaluated by examining associations between stigma scores and hypothesised correlates (PHQ-8 scores and disclosure concerns) in the full UK sample as well as the clinically diagnosed and undiagnosed/unsure sub-groups.

**Phase 2: Estimating the burden of stigma experienced by people living with Long Covid.** Two prevalence estimates were calculated. The first estimated prevalence of respondents who answered at least sometimes to one or more individual question within the overall stigma scale, and each sub-scale. The second estimated prevalence of respondents who answered often or always to one or more question within the overall scale and each sub-scale. These were also presented stratified by whether participants reported having a clinical diagnosis or not.

## Results

### Sample characteristics

A total of 1166 people completed the follow-up survey, of which 966 were from the UK. 888 responded to the LCSS questions. UK sample characteristics are described in Table 1 and full sample characteristics in S1 Table. The mean age of respondents was 48.3 (SD: 10.7), and 84.6% identified as female. The majority were living in England (81.4%), followed by Scotland (11.5%), Wales (6.1%) and Northern Ireland (0.9%). 75.5% of respondents were educated to university level or higher. Roughly half of respondents (n = 460, 50.4%) reported having a clinical diagnosis of Long Covid. 557 respondents (60.9%) said that they are very careful who they tell they have Long Covid at least 'sometimes', and 308 (33.7%) said that they regretted having told some people that they have Long Covid at least 'sometimes'. Missing data were below 10% for all items, so cases were excluded list wise [36].

### CFA results

Table 2 summarises item phrasing, response options and frequencies for stigma domains indicators that were included in the CFA. CFA was run on a 3-factor robust maximum likelihood (MLR) model. Enacted stigma items were constrained to load onto the enacted stigma factor;

**Table 1. UK sample characteristics (n = 966).**

| | n | % |
|---|---|---|
| **Age (mean 48.3, SD 10.7)** | | |
| 18–30 | 52 | 5.4 |
| 31–45 | 321 | 33.5 |
| 46–59 | 448 | 46.7 |
| ≥60 | 138 | 14.4 |
| Missing | 7 | 0.7 |
| **Gender** | | |
| Male | 139 | 14.5 |
| Female | 811 | 84.6 |
| Non-binary or other | 9 | 0.9 |
| Missing | 7 | 0.7 |
| **Ethnicity** | | |
| White | 914 | 95.5 |
| Minority ethnic | 43 | 4.5 |
| Missing | 9 | 0.9 |
| **Country of residence** | | |
| England | 778 | 81.4 |
| Scotland | 111 | 11.6 |
| Wales | 58 | 6.1 |
| Northern Ireland | 9 | 0.9 |
| Missing | 10 | 1 |
| **Educational qualification** | | |
| No formal qualifications | 10 | 1.0 |
| O levels or equivalent | 95 | 9.8 |
| A levels or equivalent | 131 | 13.6 |
| University degree or above | 729 | 75.5 |
| Other | 1 | 0.1 |
| **Duration of Long Covid illness** | | |
| <12 months | 30 | 3.1 |
| 12-<15 months | 53 | 5.6 |
| 15-<18 months | 31 | 3.3 |
| >18 months | 840 | 88.1 |
| Missing | 12 | 1.2 |
| **Employment status** | | |
| Employed | 653 | 65.6 |
| Unable to work | 281 | 29.1 |
| Student/Volunteer | 18 | 1.9 |
| Unemployed and looking for work | 14 | 1.5 |
| **Clinical diagnosis of Long Covid received or on health record** | | |
| No | 39 | 4.3 |
| Not sure | 121 | 13.3 |
| Have test confirmation of initial Covid infection but no/not sure clinical diagnosis of Long Covid | 53 | 5.8 |
| No official diagnosis but doctors suspect I have Long Covid | 240 | 26.3 |
| Yes, Long Covid as a diagnosis on health record | 460 | 50.4 |
| Missing | 53 | 5.5 |
| **Long Covid Stigma Scale (LCSS) score**, mean (SD) | 20.4 ± 10.8 | |
| Missing | 78 | 8.1 |

*(Continued)*

**Table 1.** (Continued)

|  | n | % |
|---|---|---|
| **Disclosure concerns score**, mean (SD) | 2.9 ± 2.3 | |
| Missing | 53 | 5.5 |
| **I am very careful who I tell that I have Long Covid** | | |
| At least 'sometimes' | 557 | 60.9 |
| Often or always | 296 | 32.3 |
| **I regret having told some people that I have Long Covid** | | |
| At least 'sometimes' | 308 | 33.7 |
| Often or always | 117 | 12.8 |
| **PHQ-8 score**, mean (SD) | 9.2 ± 5.8 | |
| Missing | 85 | 8.8 |

anticipated stigma items were constrained to load onto the anticipated stigma factor; and internalised stigma items were constrained to load onto the internalised stigma factor. Error terms were only allowed to be correlated between items on the same subscales.

Results of the CFA are presented in Table 3. Fit indices indicated that the model fitted the data well for the full UK sample, as well as both subsamples. Standardised factor loadings of

**Table 2. Response option frequencies for each stigma item.**

| Response options* | Full UK sample (n = 888) | | | | | Clinical diagnosis (n = 440) | | | | | No clinical diagnosis/ unsure (n = 443) | | | | |
|---|---|---|---|---|---|---|---|---|---|---|---|---|---|---|---|
| | 0 | 1 | 2 | 3 | 4 | 0 | 1 | 2 | 3 | 4 | 0 | 1 | 2 | 3 | 4 |
| **Enacted stigma items** | | | | | | | | | | | | | | | |
| Because of my illness, some people seemed uncomfortable with me | 27.5 | 20.7 | 37.4 | 13.3 | 1.1 | 18.6 | 21.4 | 43.9 | 14.8 | 1.4 | 36.1 | 20.1 | 31.4 | 11.5 | 0.9 |
| Because of my illness, some people were unkind to me | 50.0 | 21.0 | 22.8 | 6.2 | 0.1 | 43.0 | 23.2 | 27.5 | 6.4 | - | 56.9 | 18.7 | 18.3 | 5.9 | 0.2 |
| People I care about stopped contacting me after learning I have Long Covid | 55.0 | 15.7 | 19.7 | 8.9 | 0.8 | 43.0 | 18.4 | 26.1 | 12.1 | 0.5 | 66.8 | 13.1 | 13.3 | 5.6 | 1.1 |
| People have acted as if I am dishonest since I have had Long Covid | 47.3 | 19.7 | 22.2 | 9.4 | 1.5 | 43.4 | 20.9 | 23.9 | 10.9 | 0.9 | 51.2 | 18.5 | 20.5 | 7.7 | 2.0 |
| I have been treated with less respect than other people are because of Long Covid | 48.7 | 20.4 | 20.8 | 9.2 | 0.9 | 40.0 | 22.7 | 24.1 | 11.8 | 1.4 | 56.9 | 18.3 | 17.6 | 6.8 | 0.5 |
| **Internalised stigma items** | | | | | | | | | | | | | | | |
| I have felt embarrassed about my illness | 25.9 | 11.6 | 31.3 | 22.9 | 8.3 | 19.8 | 11.4 | 32.1 | 26.8 | 10.0 | 31.6 | 12.0 | 30.9 | 19.0 | 6.6 |
| I have felt embarrassed because of my physical limitations | 15.1 | 9.1 | 29.4 | 32.2 | 14.2 | 9.8 | 7.5 | 28.0 | 37.5 | 17.3 | 20.1 | 10.8 | 30.9 | 27.3 | 10.8 |
| I feel that I have been tainted by Long Covid and am of less value than others because of it | 26.8 | 15.7 | 26.6 | 19.6 | 11.4 | 16.8 | 15.7 | 28.2 | 24.6 | 14.8 | 36.3 | 15.8 | 25.3 | 14.9 | 7.7 |
| I have felt like I am very different from other people on account of Long Covid | 17.8 | 14.6 | 32.7 | 22.0 | 13.0 | 11.1 | 13.2 | 31.1 | 28.6 | 15.9 | 24.4 | 15.8 | 34.3 | 15.6 | 9.9 |
| **Anticipated stigma items** | | | | | | | | | | | | | | | |
| Many people tend to think Long Covid isn't a real illness | 7.4 | 12.5 | 37.2 | 33.0 | 9.9 | 7.1 | 14.1 | 37.5 | 29.8 | 11.6 | 7.9 | 11.1 | 36.8 | 35.9 | 8.4 |
| I feel that some people assume that having Long Covid is a sign of personal weakness | 18.0 | 15.9 | 34.2 | 24.6 | 7.3 | 11.6 | 14.8 | 37.7 | 27.1 | 8.9 | 24.4 | 16.7 | 30.7 | 22.4 | 5.9 |
| I worry that people with Long Covid lose their jobs when their employers find out | 19.9 | 14.1 | 36.0 | 22.4 | 7.6 | 12.7 | 12.5 | 38.2 | 27.3 | 9.3 | 26.9 | 15.6 | 34.3 | 17.6 | 5.6 |
| I worry that people may judge me negatively when they learn I have Long Covid | 21.3 | 16.9 | 33.5 | 19.5 | 8.9 | 13.4 | 15.7 | 35.5 | 23.2 | 12.3 | 29.1 | 18.1 | 31.6 | 15.6 | 5.6 |

*Response options indicate 0:Never; 1:Rarely; 2:Sometimes; 3:Often; 4:Always.

**Table 3. Factor loadings of individual stigma items on subscales of internalised, enacted and anticipated stigma using confirmatory factor analysis.**

| | Full UK sample (n = 888) | | | Clinical diagnosis (n = 440) | | | No clinical diagnosis/unsure (n = 443) | | |
|---|---|---|---|---|---|---|---|---|---|
| | Enacted | Internalised | Anticipated | Enacted | Internalised | Anticipated | Enacted | Internalised | Anticipated |
| Because of my illness, some people seemed uncomfortable with me | 0.78 | | | 0.74 | | | 0.78 | | |
| Because of my illness, some people were unkind to me | 0.79 | | | 0.81 | | | 0.77 | | |
| People I care about stopped contacting me after learning I have Long Covid | 0.70 | | | 0.66 | | | 0.71 | | |
| People have acted as if I am dishonest since I have had Long Covid | 0.76 | | | 0.77 | | | 0.79 | | |
| I have been treated with less respect than other people are because of Long Covid | 0.86 | | | 0.86 | | | 0.85 | | |
| I have felt embarrassed about my illness | | 0.78 | | | 0.74 | | | 0.81 | |
| I have felt embarrassed because of my physical limitations | | 0.77 | | | 0.73 | | | 0.79 | |
| I feel that I have been tainted by Long Covid and am of less value than others because of it | | 0.84 | | | 0.82 | | | 0.83 | |
| I have felt like I am very different from other people on account of Long Covid | | 0.75 | | | 0.69 | | | 0.78 | |
| Many people tend to think Long Covid isn't a real illness | | | 0.65 | | | 0.70 | | | 0.66 |
| I feel that some people assume that having Long Covid is a sign of personal weakness | | | 0.78 | | | 0.78 | | | 0.77 |
| I worry that people with Long Covid lose their jobs when their employers find out | | | 0.58 | | | 0.52 | | | 0.59 |
| I worry that people may judge me negatively when they learn I have Long Covid | | | 0.87 | | | 0.84 | | | 0.87 |
| CFI | | 0.971 | | | 0.972 | | | 0.971 | |
| TLI | | 0.958 | | | 0.959 | | | 0.958 | |
| RMSEA | | 0.064 | | | 0.062 | | | 0.064 | |
| SRMR | | 0.037 | | | 0.040 | | | 0.039 | |
| $\chi2$/df | | 4.7 | | | 2.7 | | | 2.8 | |
| Cronbach's alpha | 0.88 | 0.86 | 0.82 | 0.88 | 0.85 | 0.81 | 0.89 | 0.87 | 0.83 |

indicators onto the latent constructs were high for all three domains, ranging between 0.70–0.86 for enacted, 0.75–0.84 for internalised, and 0.58–0.87 for anticipated stigma in the full UK sample.

Latent correlations between internalised and anticipated ($r = 0.65$, $p<0.001$) and enacted stigma ($r = 0.57$, $p<0.001$) were statistically significant. Anticipated and enacted stigma were also correlated ($r = 0.67$, $p<0.001$). Modifications to the measurement model were not necessary due to the good fit and factor loadings. Cronbach's α were 0.82, 0.88 and 0.86 for anticipated, enacted and internalised stigma respectively.

Correlations testing concurrent criterion validity confirmed hypothesised relationships: the overall LCSS and the enacted, internalised and anticipated stigma subscales were consistently positively associated with PHQ-8 score and disclosure concerns (Table 4).

## Prevalence of stigma

Two prevalence estimates are presented (Table 5). Based on the first estimate, prevalence of people experiencing overall stigma at least 'sometimes' was 95.4%; prevalence of enacted

**Table 4. Correlations between stigma scores, eight-item Patient Health Questionnaire (PHQ-8 score) and disclosure concerns.**

| | Full UK sample | | | | Clinical diagnosis | | | | No clinical diagnosis/unsure | | | |
|---|---|---|---|---|---|---|---|---|---|---|---|---|
| | PHQ-8 score | p-value | Disclosure concerns | p-value* | PHQ-8 score | p-value | Disclosure concerns | p-value* | PHQ-8 score | p-value | Disclosure concerns | p-value* |
| Overall LCSS | 0.47 | <0.001 | 0.61 | <0.001 | 0.46 | <0.001 | 0.64 | <0.001 | 0.46 | <0.001 | 0.61 | <0.001 |
| Enacted stigma subscale | 0.35 | <0.001 | 0.50 | <0.001 | 0.32 | <0.001 | 0.54 | <0.001 | 0.32 | <0.001 | 0.47 | <0.001 |
| Internalised stigma subscale | 0.48 | <0.001 | 0.52 | <0.001 | 0.47 | <0.001 | 0.53 | <0.001 | 0.47 | <0.001 | 0.51 | <0.001 |
| Anticipated stigma subscale | 0.39 | <0.001 | 0.59 | <0.001 | 0.40 | <0.001 | 0.59 | <0.001 | 0.40 | <0.001 | 0.59 | <0.001 |

*Comparisons used Pearson's correlation.

stigma was 62.7%; internalised stigma was 86.4%; and anticipated stigma was 90.8%. According to the more conservative estimate 'often or always', prevalence of overall, enacted, internalised and anticipated stigma experienced 'often' or 'always' was 75.9%, 25.3%, 59.7%, and 59.0% respectively. For all types of stigma and using both estimates, those with a clinical diagnosis of Long Covid had a higher prevalence than those without (Fig 1).

## Discussion

### Summary

This paper describes the development and validation of the first psychometric scale to measure stigma associated with Long Covid and offers the first quantitative estimate of the burden within a UK sample. The new scale captures three key domains—enacted, internalised and anticipated stigma. It demonstrated good psychometric properties within the overall sample, and sub-samples of those with and without Long Covid diagnoses.

Prevalence estimates using this new validated scale suggest that the majority of people with Long Covid are experiencing some form of stigma, with 95.4% experiencing at least one type at least 'sometimes', and 75.9% experiencing it 'often'. Anticipated and internalised stigma were more frequently experienced than enacted stigma, in line with evidence of stigma associated with other concealable conditions [37, 38].

Prevalence of stigma was higher in those who reported having a clinical diagnosis of Long Covid. The reason is not clear. It may be that this group were exposed to more stereotyping or dismissal of their experience during their journey to obtaining a clinical diagnosis compared

**Table 5. Prevalence of stigma in the UK-based sample.**

| | Experienced stigma at least 'sometimes' | | | | Experienced stigma often/always | | | |
|---|---|---|---|---|---|---|---|---|
| | Full UK sample | Clinical diagnosis | No clinical diagnosis/ unsure | p-value* | Full UK sample | Clinical diagnosis | No clinical diagnosis/ unsure | p-value* |
| Overall LCSS | 95.4 | 97.5 | 93.2 | 0.02 | 75.9 | 82.5 | 69.3 | <0.001 |
| Enacted stigma | 62.7 | 70.7 | 55.1 | <0.001 | 25.3 | 29.3 | 21.2 | 0.007 |
| Internalised stigma | 86.4 | 91.8 | 81.3 | <0.001 | 59.7 | 70.0 | 49.4 | <0.001 |
| Anticipated stigma | 90.8 | 93.2 | 88.3 | 0.007 | 59.0 | 63.6 | 54.6 | 0.004 |

*Comparisons between those with a clinical diagnosis and those with no clinical diagnosis/unsure used chi square test.

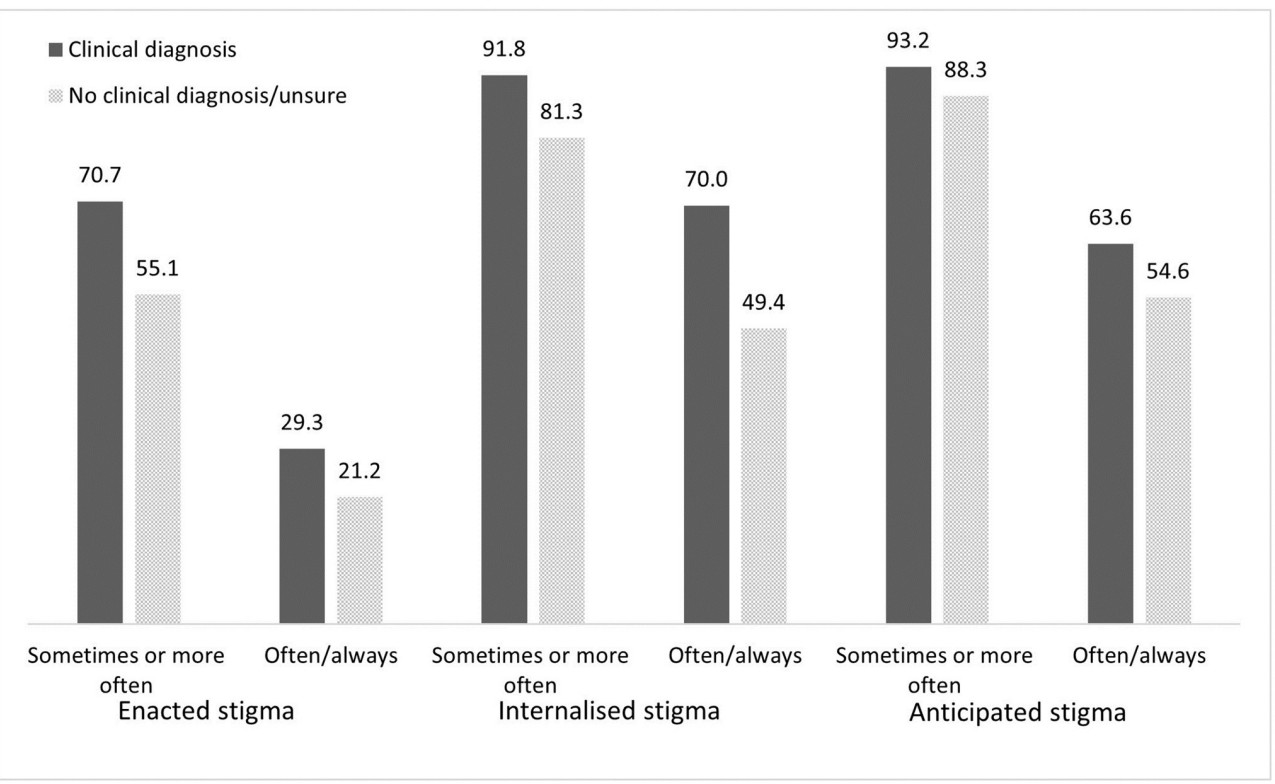

**Fig 1. Prevalence estimates of stigma among people with diagnosed and undiagnosed Long Covid.**

to those with no clinical diagnosis who perhaps had less interaction with services and others about their Long Covid. It may be that their Long Covid is more severe in nature making it more visible to others and/or more impactful in limiting everyday activities.

## Strengths and limitations

LCSS was informed by theory and other stigma scales, co-designed with people with Long Covid and validated in a large UK sample, and takes less than 10 minutes to complete. This study has two notable limitations. First, the convenience non-probability sampling limits generalisability: university-educated white women from England are over-represented, and this may have resulted in an under- or over-estimation of Long Covid stigma. The prevalence of Long Covid in the UK is higher in females and in those aged 35–69 years [39]. The latest estimates from the UK's Office of National Statistics also point to a higher prevalence in people living in more deprived areas, those working in social care, those aged 16 or over who are not in employment, and those with another activity-limiting health conditions [40]. Women and ethnic minorities may be more stigmatised by other similar conditions such as ME/CFS or fibromyalgia [41], though there is little quantitative evidence to support this. The survey did not include patients hospitalised with COVID-19 in the first two weeks of illness, indicating severe acute disease. Stigma levels could be higher in this group as they may have a higher prevalence of prolonged ongoing symptoms [42], or could be lower due to legitimisation of their illness given by the severity of its acute stage.

However, the community sample renders this study unique within a largely clinical evidence-base where diagnostic coding for Long Covid remains patchy and inconsistent due to varying knowledge and the absence of specific guidelines [43]. The social media recruitment

strategy aimed to include an underrepresented group of people living with Long Covid–those not actively engaged with the healthcare system.

Second, stigma is a non-pathological construct and measurements do not have standardised diagnostic criteria. The aim is to capture frequency of specific experiences, and there is no agreed cut-off for measuring prevalence. This study sought to overcome potential bias by offering two different estimates, following cut-offs used in other studies on health-related stigma [44, 45]. This has resulted in divergent estimates capturing the percentage of people who experience different stigma mechanisms at least 'sometimes', and those who experience it often/always.

## Comparison with existing literature

The new LCSS is an important addition to the growing number of existing scales for measuring stigma among people with acute COVID-19 infection [46–48] or at risk of such infection [49]. The stigma associated with Long Covid is unique, requiring its own measurement tool; unlike acute COVID-19, Long Covid is a chronic, less well understood condition, often psychologised, and has serious implications for people's long-term health and productivity.

Existing evidence on Long Covid stigma has been largely qualitative and anecdotal [8–11], which has been essential for describing and raising awareness around the issue. This study adds to ongoing narratives by quantifying the burden of stigma experienced by people living with Long Covid.

## Implications for research and/or practice

Findings highlight widespread and multi-layered stigmas experienced by people living with Long Covid in the UK, which should be taken into consideration within clinical practice and healthcare policy. Whilst education about Long Covid may be an important first step, it is not a magic bullet for addressing stigma [50]. Evidence from other stigmatised health conditions suggests that interventions that facilitate social contact with the stigmatised group, advocacy and community mobilisation, as well as peer services may reduce both stigmatising attitudes and internalised stigma [51, 52]. Peer service providers would be uniquely positioned to foster non-judgmental environments and accountability to patients within Long Covid clinics and general practice.

Evidence from across health conditions and geographic contexts suggests Long Covid stigma could be hindering public health by compromising patients' mental health and engagement with the health system. Developing evidence-based strategies to tackle Long Covid stigma requires a description of the problem, including prevalence estimates, and a validated scale that can capture changes in stigma over time. It is hoped that this study will enable further research on predictors of and interventions to address Long Covid stigma, including further exploring how the prevalence of such stigma differs by social and demographic factors. This would require efforts to recruit a more representative study sample with regards to age, ethnicity, profession and socioeconomic status. Adding measurement of stigma to the core outcomes recommended for Long Covid research would allow a more comprehensive assessment of the problem and provide insights on how to improve patient outcomes and reduce inequalities. More research on how stigma plays a role in changing the social identity of people with Long Covid is also needed.

## Supporting information

**S1 Table. Sample characteristics (n = 1166).**
(DOCX)

**S2 Table. Response option frequencies for each stigma item.**
(DOCX)

**S3 Table. Factor loadings of individual stigma items on subscales of internalised, enacted and anticipated stigma using confirmatory factor analysis.**
(DOCX)

**S4 Table. Correlations between stigma scores, eight-item Patient Health Questionnaire (PHQ-8 score) and disclosure concerns.**
(DOCX)

**S5 Table. Prevalence of reported stigma.**
(DOCX)

## Acknowledgments

We thank all participants for their time and commitment completing this survey. We also sincerely thank members of Long Covid Support's COVID-19 Research Involvement Group for providing feedback on earlier versions of the questionnaire.

## Author Contributions

**Conceptualization:** Marija Pantelic, Nisreen A. Alwan.

**Data curation:** Nida Ziauddeen.

**Formal analysis:** Nida Ziauddeen.

**Methodology:** Marija Pantelic, Nida Ziauddeen, Mark Boyes, Margaret E. O'Hara, Claire Hastie, Nisreen A. Alwan.

**Project administration:** Nida Ziauddeen.

**Writing – original draft:** Marija Pantelic, Nida Ziauddeen, Nisreen A. Alwan.

**Writing – review & editing:** Marija Pantelic, Nida Ziauddeen, Mark Boyes, Margaret E. O'Hara, Claire Hastie, Nisreen A. Alwan.

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
