## [Decision Letter · Decision Letter 0]

15 Aug 2022

PONE-D-22-19124Long Covid stigma: estimating burden and validating scale in a UK-based samplePLOS ONE

Dear Dr. Ziauddeen,

Thank you for submitting your manuscript to PLOS ONE. After careful consideration, we feel that it has merit but does not fully meet PLOS ONE’s publication criteria as it currently stands. Therefore, we invite you to submit a revised version of the manuscript that addresses the points raised during the review process.

We look forward to receiving your revised manuscript.

Kind regards,

Dong Keon Yon, MD, FACAAI

Academic Editor

PLOS ONE

Journal Requirements:

Additional Editor Comments:

I read it with great interest. Please address excellent comments from the reviewers.

Reviewers' comments:

Reviewer's Responses to Questions

**Comments to the Author**

1. Is the manuscript technically sound, and do the data support the conclusions?

Reviewer #1: Partly

Reviewer #2: Yes

Reviewer #3: Yes

2. Has the statistical analysis been performed appropriately and rigorously? 

Reviewer #1: I Don't Know

Reviewer #2: Yes

Reviewer #3: Yes

3. Have the authors made all data underlying the findings in their manuscript fully available?

Reviewer #1: Yes

Reviewer #2: Yes

Reviewer #3: No

4. Is the manuscript presented in an intelligible fashion and written in standard English?

Reviewer #1: Yes

Reviewer #2: Yes

Reviewer #3: Yes

5. Review Comments to the Author

Reviewer #1: The study is very relevant . My only concern in with the use of sometimes/often etc as a scale. As a scale I think you move from 'sometimes' to 'often' to 'more often', then 'always'. I therefore think the authors should use 'sometime/often' and 'more often/always' as opposed to 'sometimes/more often' and 'often/always.

For the rest methods and results, I do not have adequate statistical expertise to evaluate.

Reviewer #2: Manuscript has been written in an intelligible fashion and in depth statistical analysis has been performed.It presented with a good flow and very understandable manner. It is very useful study in the present context. I congratulate all authors for their heard work.

Reviewer #3: Overall, an interesting read and very well written. Would like to see a bit more discussion on this limitation: university educated white women from England are over-represented. Do you have any information on the prevalence of Long Covid in the UK and who is most impacted? Men or women? Socioeconomic status? Age? Etc. I don't have a point of comparison for understanding the results. Also would the next steps be to get a more generalizable sample across SES? It may be that your results will be different with a more diverse sample. An interesting future article would be to look at how stigma differs by gender, SES, age, etc.

6. PLOS authors have the option to publish the peer review history of their article (what does this mean?). If published, this will include your full peer review and any attached files.

Reviewer #1: No

Reviewer #2: No

Reviewer #3: No

---

## [Author Response · Author response to Decision Letter 0]

27 Sep 2022

Thank you, we have formatted the manuscript in line with PLOS ONE’s style requirements.

Written consent was recorded digitally on the survey platforms (details in lines 93-95). The survey was restricted to adults aged 18 years and over (line 84).

This study received no specific funding as stated in the funding information and financial disclosure sections. The funding information in the manuscript has been updated so the sections match.

Additional Editor Comments:

I read it with great interest. Please address excellent comments from the reviewers.

Thank you. 

Review Comments to the Author

Reviewer #1: The study is very relevant . My only concern in with the use of sometimes/often etc as a scale. As a scale I think you move from 'sometimes' to 'often' to 'more often', then 'always'. I therefore think the authors should use 'sometime/often' and 'more often/always' as opposed to 'sometimes/more often' and 'often/always.

The possible responses to the scale were never, rarely, sometimes, often, and always. We created two condensed categories: 1) those who responded sometimes/often/always to reflect the proportion of those experiencing stigma at least sometimes (referred to as ‘sometimes or more often’ in the tables) and 2) a more conservative estimate of those who responded often/always to reflect the proportion of those experiencing some stigma more frequently (‘referred to often and always’ in the tables). For the avoidance of confusion, we have now updated any reference to the former to ‘at least sometimes’ throughout the manuscript and have kept the description of the latter to ‘often or always’

For the rest methods and results, I do not have adequate statistical expertise to evaluate.

Reviewer #2: Manuscript has been written in an intelligible fashion and in depth statistical analysis has been performed. It presented with a good flow and very understandable manner. It is very useful study in the present context. I congratulate all authors for their heard work.

Thank you. 

Reviewer #3: Overall, an interesting read and very well written. Would like to see a bit more discussion on this limitation: university educated white women from England are over-represented. Do you have any information on the prevalence of Long Covid in the UK and who is most impacted? Men or women? Socioeconomic status? Age? Etc. I don't have a point of comparison for understanding the results. Also would the next steps be to get a more generalizable sample across SES? It may be that your results will be different with a more diverse sample. An interesting future article would be to look at how stigma differs by gender, SES, age, etc.

Thank you, we have added more information in the discussion section on prevalence rates in lines (218-222). In the UK, prevalence of Long Covid in higher in women and in adults aged 35-69 years and in more deprived communities. 

We agree with the reviewer’s suggestions about future research directions on this topic and have added this to the discussion (lines 262-264). We thank the reviewer for suggesting these thoughtful discussion points.

---

## [Decision Letter · Decision Letter 1]

25 Oct 2022

Long Covid stigma: estimating burden and validating scale in a UK-based sample

PONE-D-22-19124R1

Dear Dr. Ziauddeen,

We’re pleased to inform you that your manuscript has been judged scientifically suitable for publication and will be formally accepted for publication once it meets all outstanding technical requirements.

Kind regards,

Dong Keon Yon, MD, FACAAI

Academic Editor

PLOS ONE

Additional Editor Comments (optional):

This is an excellent paper.

Reviewers' comments:

Reviewer's Responses to Questions

**Comments to the Author**

1. If the authors have adequately addressed your comments raised in a previous round of review and you feel that this manuscript is now acceptable for publication, you may indicate that here to bypass the “Comments to the Author” section, enter your conflict of interest statement in the “Confidential to Editor” section, and submit your "Accept" recommendation.

Reviewer #2: All comments have been addressed

Reviewer #3: All comments have been addressed

2. Is the manuscript technically sound, and do the data support the conclusions?

Reviewer #2: Yes

Reviewer #3: Yes

3. Has the statistical analysis been performed appropriately and rigorously? 

Reviewer #2: Yes

Reviewer #3: Yes

4. Have the authors made all data underlying the findings in their manuscript fully available?

Reviewer #2: Yes

Reviewer #3: No

5. Is the manuscript presented in an intelligible fashion and written in standard English?

Reviewer #2: Yes

Reviewer #3: Yes

6. Review Comments to the Author

Reviewer #2: I thank all author for their wounder full work. Authors have address all the review comments and re arrange the manuscript. It is a very interesting study.

Reviewer #3: (No Response)

7. PLOS authors have the option to publish the peer review history of their article (what does this mean?). If published, this will include your full peer review and any attached files.

Reviewer #2: No

Reviewer #3: No
